# Mitigating Framing Bias with Polarity Minimization Loss

**Yejin Bang**    **Nayeon Lee**    **Pascale Fung**
Centre for Artificial Intelligence Research (CAiRE)
The Hong Kong University of Science and Technology
yjbang@connect.ust.hk

## Abstract

Framing bias plays a significant role in exacerbating political polarization by distorting the perception of actual events. Media outlets with divergent political stances often use polarized language in their reporting of the same event. We propose a new loss function that encourages the model to minimize the polarity difference between the polarized input articles to reduce framing bias. Specifically, our loss is designed to jointly optimize the model to map polarity ends bidirectionally. Our experimental results demonstrate that incorporating the proposed polarity minimization loss leads to a substantial reduction in framing bias when compared to a BART-based multi-document summarization model. Notably, we find that the effectiveness of this approach is most pronounced when the model is trained to minimize the polarity loss associated with informational framing bias (i.e., skewed selection of information to report).

## 1 Introduction

Framing bias has become a pervasive problem in modern media, misleading the understanding of what really happened via a skewed selection of information and language (Entman, 2007, 2010; Gentzkow and Shapiro, 2006). The most notable impact of framing bias is the amplified polarity between conflicting political parties and media outlets. Mitigating framing bias is critical to promote accurate and objective delivery of information.

One promising mitigation paradigm is to generate a neutralized version of a news article by synthesizing multiple views from biased source articles (Sides, 2018; Lee et al., 2022). To more effectively achieve news neutralization, we introduce a polarity minimization loss that leverages inductive bias that encourages the model to prefer generation with minimized polarity difference. Our proposed loss trains the model to be simultaneously good at mapping articles from one end of the polarity spectrum to another end of the spectrum and vice versa as

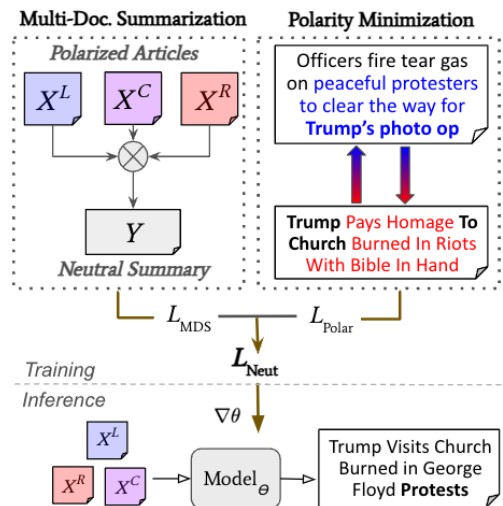

Figure 1: Illustration of training and inference with the proposed polarity minimization loss for reducing framing bias.

illustrated in Fig. 1. Intuitively, the model is forced to learn and focus on the shared aspect between contrasting polarities from two opposite ends.

In this work, we demonstrate the effectiveness of our proposed loss function by minimizing polarity in different dimensions of framing bias – lexical and informational (Entman, 2002). Lexical polarization results from the choice of words with different valence and arousal to explain the same information (e.g., "protest" vs "riot"). Informational polarization results from a differing selection of information to cover, often including unnecessary or unrelated information related to the issue being covered. Our investigation suggests that learning the opposite polarities that are distinct in the informational dimension enables the model to acquire a better ability to focus on common ground and minimize biases in the polarized input articles. Ultimately, our proposed loss enables the removal of bias-inducing information and the generation of more neutral language choices.

## 2 Related Work

**Framing Bias** Framing bias is a well-documented phenomenon in the field of media studies (Wright and Goodwin, 2002; Entman, 2002, 2010, 2007; Gentzkow and Shapiro, 2006; Gentzkow et al., 2015; Beratšová et al., 2016). According to Gentzkow and Shapiro (2006), framing bias occurs when journalists and media outlets selectively emphasize certain aspects of a story while downplaying or ignoring others (informational) with biased use of languages (lexical). This can result in a distorted perception of events among the public, particularly in cases where the framing is done to serve a particular agenda or ideology (Kahneman and Tversky, 2013; Goffman, 1974). The impact of framing bias is especially evident in the political arena, where media outlets and political parties often engage in polarizing discourse that is designed to appeal to their respective bases (Scheufele, 2000; Chong and Druckman, 2007).

**Automatic Mitigation Efforts** To mitigate that, there have been various automatic media bias mitigation efforts (Fan et al., 2019; Hamborg et al., 2019; Morstatter et al., 2018; Laban and Hearst, 2017; Hamborg et al., 2017; Zhang et al., 2019b; van den Berg and Markert, 2020; Lee et al., 2022). A similar line of work is ideology prediction (Liu et al., 2022) (if they are left-, right-, or center-leaning) or stance prediction (Baly et al., 2020) – which is polarity detection. On the other hand, our work focuses on generating a neural article from polarized articles. Given that framing bias often happens very subtle, Morstatter et al. (2018) learns the pattern of framing bias in a sentence and attempts to detect it automatically. Another common mitigation attempt is to display multiple viewpoints in an automatic way (Hamborg et al., 2017; Park et al., 2009). Lee et al. (2022) took a further step to make a summary out of the polarized articles to provide multiple perspectives automatically in one single summary. Our work aligns with the vision of previous works, but we focus on the more general way to mitigate framing bias by studying the polarity minimization loss.

## 3 Approach

### 3.1 Problem Statement

Given a set of multiple polarized news articles $X^{1...K}$ with varying degrees and orientations of political bias, the goal is to generate a neutral article summary $Y$, where dataset is $D = (\{X^1, X^2, \ldots, X^K\}, Y)$. The neutral summary $Y$ should (i) retain salient information and (ii) minimize as much framing bias as possible from the input articles.

### 3.2 Polarity Regularized Model

To equip a model for neutral summarization, we formulate it as a conditional generation with given polarized articles $X^{polarized} = \{X^1, X^2, \ldots, X^K\}$, where superscripts represent arbitrary polarization directions based in a certain criteria dimension. In this work, it is based on political ideologies (i.e., left ($X^L$), right ($X^R$), center ($X^C$)). The model can be defined in a conditional auto-regressive manner as defined in eq. 2. Input $X$, defined in eq. 1, is yielded to vector representations by the encoder. Then the encoded input polarized articles are used to generate a target neutral summary sequentially using the decoder.

$$X = \mathtt{concat}((X^i : i \in \mathtt{shuffle}([K]))), \quad (1)$$

where $\mathtt{shuffle}(l)$ denotes the random permutation operation on a given list $l$.

$$p_\theta(Y|X) = \prod_{t=1}^{V} p_\theta(y_v|y_{<v}, X), \quad (2)$$

where $V$ denotes the length of the article summary $Y$. The model parameterized by $\theta$ is optimized through the objective maximum likelihood of target tokens in the neutral article $Y$. Precisely, the loss function for training this model is as follows:

$$L_{\mathrm{MDS}} = -\frac{1}{|B|} \sum_{X,Y \in B} \sum_{t=1}^{T} \log p_\theta(y_t|y_{<t}, X) \quad (3)$$

where $B$ is a batch of input and target pairs $(X, Y)$. However, $L_{MDS}$ does not explicitly optimize for polarity distance.

**Polarity Minimization Loss** Framing bias results from the polarized portrayal of the same event or issue. Motivated by this, we propose to train the model with an additional polarity minimization loss, $L_{\mathrm{polar}}$, which requires learning from both directions between arbitrary polarities (e.g., $X^1 \rightarrow X^K; X^K \rightarrow X^1$). Given there are two polarity ends, an article with arbitrary source polarity is denoted as $X^s$ while an opposite polarity is a target article, $X^t$. The polarity minimization loss

| | Avg. Framing Bias Metric ↓ | | | Salient Info ↑ | |
|---|---|---|---|---|---|
| | $Arousal_+$ | $Arousal_-$ | $Arousal_{sum}$ | BLEU | BERTS-F1 |
| All Source Input | 6.76 | 3.64 | 10.4 | 8.27 | 64.00% |
| PEGASUSMULTI | 5.12 | 2.39 | 7.51 | 6.12 | 61.44% |
| BARTMULTI | 5.94 | 2.66 | 8.61 | 4.24 | 61.06% |
| PEGASUSNEUSFT | 2.18 | 1.12 | 3.30 | 11.26 | 68.41% |
| BARTNEUSFT | 1.86 | 1.00 | 2.85 | 11.67 | 70.10% |
| BARTNEUSFT-T | 1.69 | 0.83 | 2.53 | 12.05 | 70.50% |
| +LR-VALENCE | 1.57 | 0.91 | 2.48 | **10.62** | 69.67% |
| +LR-AROUSAL | 1.18 | **0.62** | 1.80 | 8.84 | 69.49% |
| +LR-INFO | **1.08** | 0.70 | **1.78** | 9.31 | 70.19% |
| +LRC-AROUSAL | 1.22 | 0.75 | 1.97 | 9.66 | 69.94% |
| +LRC-INFO | 1.25 | 0.72 | 1.97 | 10.18 | **70.24%** |

Table 1: Experimental results for ALLSIDES test set. For framing bias metric, the *lower* number is the better (↓). For other scores, the *higher* number is the better (↑).

obtained with negative log-likelihood between one polarity $X^s$ to the opposite $X^t$ is as follows:

$$L_{\text{polar}} = -\frac{1}{|B|} \sum_{X^s, X^t \in B} \sum_{i=1}^{|X^t|} \log p_\theta(x_i | x_{<i}, X^s) \quad (4)$$

where the B represents a batch consisting of two articles of opposing polarities $X^s$ and $X^t$. Note that $s, t$ are not bound to specific polarities but indicate source and target ends (i.e., $X^t \in \{X^1, X^K\}$ and $X^s \in \{X^1, X^2, \dots, X^K\} \setminus \{X^t\}$ within a batch).

In summary, the model parameters $\theta$ are optimized with $L_{\text{neut}}$, which is composed of a multi-article summarization objective ($L_{\text{MDS}}$) and a polarity minimization term $L_{\text{polar}}$.

$$L_{\text{neut}} = L_{\text{MDS}} + \lambda \cdot L_{\text{polar}} \quad (5)$$

where $\lambda > 0$ and is a hyperparameter that assigns a relative weight to the polarity minimization loss with respect to the multi-document summarization.

## 4 Experiments

### 4.1 Setup

**Dataset** ALLSIDES dataset ($D$) (Lee et al., 2022) is from Allsides.com, designed for neutral multi-article summarization. It consists of 3066 triplets from left, right, and center American publishers on the same event, $\{X^L, X^R, X^C\}$, and an expert-written neutral summary (target) of the articles, $Y$. Note that "center" ideology contains relatively less bias, but still tends to contain framing bias (all, 2021) (Refer to Appendix B for examples).

**Metric** We evaluate with a suite of metrics introduced in the benchmark. The effectiveness of

reducing framing bias is evaluated by adopting Arousal scores (Lee et al., 2022), which is based on the Valence-Arousal-Dominance (VAD) lexicons (Mohammad, 2018). We report all positive-valence arousal scores ($Arousal_+$), negative-valence arousal scores ($Arousal_-$), and the combined arousal scores ($Arousal_{sum}$). It is ideal for models to achieve lower $Arousal_{sum}$ scores. For salient info, we mainly evaluate with BERTSCORE-F1(Zhang* et al., 2020), which is a representative embedding-based metric, instead of lexicon-based reference metrics because we expect the model to rewrite biased uses of language in a neutral way, thus, the embedding-based metric is more appropriate than the lexicon-based. For comparison with previous work, we still report BLEU (Papineni et al., 2002) score. Lastly, we conduct human-evaluation to validate the result on reducing framing bias.

### 4.2 Models

**Baselines** We compare with off-the-shelf multi-document summarization (MDS) models trained on Multi-news dataset (Fabbri et al., 2019) (BARTMULTI (Lewis et al., 2019) and PEGASUS-MULTI (Zhang et al., 2019a)) as baselines. Those models have achieved high performance in MDS, which can also be applied in summarizing polarized articles. However, these models do not have any learning about framing bias removal or neutral writing. We also compare with the state-of-the-art models (BARTNEUSFT and BARTNEUSFT-T) (Lee et al., 2022) that are fine-tuned with ALL-SIDES dataset. BARTNEUSFT is fine-tuned only with articles meanwhile BARTNEUSFT-T additionally leverages titles of each article. We additionally

| Input (X) | $\mathbf{X^L}$ : Officers fire tear gas on peaceful protesters to clear the way for Trump's photo op |
| | $\mathbf{X^R}$ : Trump Pays Homage To Church Burned In Riots With Bible In Hand |
| | $\mathbf{X^C}$ : Tear gas, threats for protesters before Trump visits church |
| Target ($Y$) | Controversy Surrounds Trump Church Visit, Dispersal of Protesters |
| Generations | **[LRC-AROUSAL]** Trump Visits Church Burned in Riots. |
| | **[LR-INFO (best) ]** Trump Visits Church Burned in George Floyd Protests. |
| | **[ −'L to R' ]** Trump Pays Homage to Church Burned in Riots. |
| | **[ −'R to L' ]** Trump Speaks on George Floyd Protests. |

Table 2: Illustration of sample generations. Our best model LR-INFO could successfully generate the neutral summary of the inputs $X^L, X^R, X^C$. −'L to R' denotes the ablation study of the effect of subtracting 'L to R' direction (i.e., having only R to L, not both) with LR-INFO.

report PEGASUSNEUSFT. Simply fine-tuning may not be effective enough to learn about framing bias. Thus, we will demonstrate how the polarity minimization loss can effectively mitigate framing bias compared to baseline and SOTA models.

**Proposed models with polarity minimization loss** The degrees of polarization vary among sets of articles and the polarization is expressed through lexical and/or informational bias. In our task, $X^s, X^t$ is in $\{X^L, X^R, X^C\}$, where L, R, and C denote left- and right-wing center political ideologies respectively. For the experiment, we used BART as the backbone model, thus our proposed models are denoted with '+' signs prepended to specific polarity minimization loss variation in Table 1. We investigate which criteria would be helpful to construct the most distinct $\{X^s, X^t\}$. Thus, variations are explained as follows:

**1) Lexical: LR-VALENCE:** Learning from the set of two extreme political ideologies articles that have a high difference in valence scores, using VAD lexicons. After picking pairs of $\{X^L, X^R\}$ articles with high differences in valence scores, we teach models about polarity shift from one another (i.e., $X^L \rightarrow X^R, X^R \rightarrow X^L$). **LR-AROUSAL:** Similar to LR-VALENCE, but based on the high differences in arousal scores. **LRC-AROUSAL:** We construct a set with all three ideologies.[1] To pick pairs of articles that have high differences in arousal score from $\{X^L, X^R, X^C\}$, we calculate differences of all combinations, which are $\{X^L, X^R\}$, $\{X^L, X^C\}$, $\{X^R, X^C\}$.

**2) Informational:** This focuses on polarization derived from "what" information to be covered. To learn the informational polarization, we select the article pairs that have high differences in information. We calculate the difference through the sum

of the number of unique tokens from each article pair. For instance, given an article pair $\{X^s, X^t\}$, we calculated as below:

$$\texttt{UniqueNum}(X^s, X^t) = |((\texttt{Token}(X^s) \cup \texttt{Token}(X^t)) \\ -(\texttt{Token}(X^s) \cap \texttt{Token}(X^t))|$$

where $\texttt{Token(X)}$ refers to a set of tokens of article X separated by blank spaces. **LR-INFO**: construct with $\{X^L, X^R\}$ **LRC-INFO**: construct from $\{X^L, X^R, X^C\}$.

### 4.3 Results

**Effectiveness of polarity minimization loss in reducing framing bias** As illustrated in Table 1, all variations of polarization minimization loss could successfully reduce overall arousal score $Arousal_{sum}$ in comparison to the baseline models. LR-INFO achieves reduction up to 8.62 absolute value compared to the bias score of All source articles (All Source), which is 82.89% reduction. Compared to baselines, our models could keep overall salient information relatively well in terms of both BLEU and BERTSCORE-F1. We investigate that loss based on informational polarization are more effective to keep the salient info than lexical polarization. For instance, $Arousal_{sum}$ are similar for LR-AROUSAL (1.80) and LR-INFO (1.78), but LR-INFO shows higher BLEU and BERTSCORE-F1– by 0.47 and 0.7% respectively.

Compared to the previous SOTA model BARTNEUSFT-T, our proposed polarity minimized models could successfully reduce overall arousal scores. Specifically, LR-INFO achives 7% of reduction. It could be achieved with relatively more reduction in $Arousal_+$(about 36% of reduction from SOTA). However, there is a bit of trade-off in salient information compared to the SOTA model ($-0.83 \sim -0.31\%$ in BERTSCORE-F1).

---

[1]Here, $X^C$ is also a biased article as explained in §4.1.

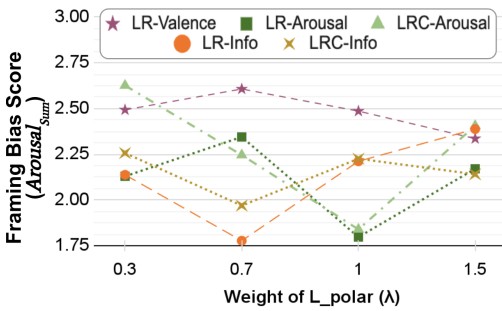

Figure 2: Illustration of performances in terms of framing bias score ($Arousal_{sum}$) depends on varying relative weights $\lambda$ for polarity minimization loss $L_{polar}$.

| | Avg. Framing Bias Metric | | | Salient Info |
|---|---|---|---|---|
| | $Arous._+$ | $Arous._-$ | $Arous._{sum}$ | BERTS-F1 |
| LR-INFO | 1.081 | 0.698 | **1.778** | 70.24% |
| − "R to L" | 1.498 | 0.830 | 2.329 | 70.06% |
| − "L to R" | 1.597 | 0.869 | 2.466 | 70.61% |

Table 3: Ablation study: Effect of having only single-directional polarity minimization with LR-INFO model.

**Effective learning with extreme polarities**  We investigate that polarity minimization between extreme ends (left, right) is more effective than the mixture with a center media outlet. This is because left and right-wing ideologies are the opposite ends that can train models more effectively about extreme ends than center media outlets although center media is not completely free of bias. Qualitative analysis results align with the quantitative measures. For instance, as illustrated in Table 2, the polarity minimized models LR-INFO and LRC-AROUSAL both could summarize with the essential information out of polarized input articles. Especially LR-INFO, the lowest biased model, it could even use a more neutral choice of word (e.g., "protests" instead of "riots" same to target $Y$).

**Human evaluation**  We conducted a human evaluation on LR-INFO against BARTNEUSFT-T, a model **without** polarity minimization loss, by asking which article is more biased. The generations from the model with LR-INFO are annotated to be less biased or similarly neutral 76.63% of the time (win: 53.3%; draw: 23.33%). Annotators agreed moderately, with Cohen's $\kappa = 0.427$ on average. According to analysis, $L_{polar}$ shows its strength to remove information that is implemented to frame in certain ways in polarized input articles. Details and examples are in Appendix A.

### 4.4   Analysis

**Variant of weights of $L_{polar}$ ($\lambda$)**   $\lambda$ is a hyper-parameter that assigns weight to the polarity minimization loss ($L_{polar}$) with respect to the multi-document summarization loss ($L_{MDS}$). $\lambda$ also indicates the intensity of the signal for learning opposite polarity. As illustrated in Fig. 2, the framing bias scores increase when polarity minimization loss overtakes MDS loss (i.e., $\lambda = 1.5$) for most

of the models. In general, models remove framing bias most effectively when $L_{polar}$ weighs the same ($\lambda = 1$) or slightly less ($\lambda = 0.7$) with respect to $L_{MDS}$. The variant indicates there exist optimal weights for each model instead of a universal optimal weight.

**Ablation Study: Polarity shift in uni-direction**  Our polarity minimization loss forces the model to learn polarity shifts bi-bidirectionally at the same time to aid in reducing framing bias. We conduct an ablation experiment of the training model with a loss optimizing polarity in uni-direction by subtracting one direction each from LR-INFO (i.e., by subtracting "R to L", we explore the effect of learning $X^L \to X^R$ only). The results support the effectiveness of our proposed bi-directional minimization loss (Table 3). This is because uni-directional learning does not lead the model to a "neutral" state of writing but to the oppositely biased polarity. In Table 2, about the issue of Trump's church visit, one direction mapping could not generate the neutral and essential information, but merely remove information that does not exist in the opposite. For instance, when subtracting mapping from L to R (i.e., − "L to R"), generation copies most from $X^R$ except, instead of minimizing the polarity difference. On the contrary, bi-directional polarity minimization loss (LR-INFO) could obtain salient information of "church visit" and neutral choice of word.

## 5   Conclusion

Framing bias is a pervasive problem in modern media, which can lead to a distorted understanding of events and an amplification of polarization. To tackle this, we introduce a polarity minimization loss that reduces framing bias in a generation. Our experimental results demonstrate that incorporating the proposed polarity minimization loss is effective in the reduction of biased uses of language and in removing biased information from the source input, which ultimately mitigates framing bias.

## Limitations

The study is limited by its adherence to the benchmark's English-based task setup. The analysis is constrained to political ideologies in the United States and the English language. Additionally, the BART model's 1024 sub-token input limit restricts the number of biased source articles that can be included as an input. It is important to note that these limitations, while potentially impacting the scope of the study's findings, are not uncommon in natural language processing research. Nonetheless, future research may benefit from addressing these limitations by exploring alternative methods for a broader range of political ideologies (Non-U.S. political ideologies) and languages, as well as incorporating longer input texts to capture a more comprehensive range of source articles.

## Ethics Statement

The issue of biased articles with framing has been extensively studied, as it can lead to polarization by influencing readers' opinions toward a certain person, group, or topic. To address this problem, our research focuses on introducing a loss function that can be incorporated to enable the model to reduce framing bias in the generated summary.

However, it is important to recognize that automatic technologies can also have unintended negative consequences if not developed with careful consideration of their broader impacts. For example, machine learning models can introduce bias in their output, replacing known source bias with another form of bias (Lee et al., 2022). To mitigate this risk, Lee et al. (2022) have suggested including explicit mention of the source articles alongside automatically generated neutral summaries. Furthermore, while our work aims to remove framing bias in human-generated articles, there is the potential for hallucination in the generation, which is a well-known problem of generative models (Ji et al., 2023). Thus, it is important to equip a guardrail (e.g., a provision of source reference) if such automatic technology is implemented for actual use cases.

Despite these challenges, our research can contribute to the effort of mitigating human-generated framing bias in order to reduce polarization in society. One of the use cases can be to aid human experts in the process of providing multi-view synthesized articles without framing bias. In terms of broader societal impact, we hope our work can help online users access more depolarized information online.

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

# A   Experimental Details

**Training Details**   Hyperparameters setup follows the benchmark for the fair comparison: $3e-5$ learning rate, batch size of 16 and 10 epochs with the early stopping of the patience of 3. Training takes around $20 \sim 30$ minutes per epoch. All experiments were with NVIDIA GTX 1080Ti GPU device. We do a hyper-parameter search for weights of the polarity minimization loss $L_{\text{polar}}$ with respect to $L_{\text{MDS}}$ in the range of $\lambda \in \{0.3, 0.7, 1, 1.5\}$.

**BERTSCORE-F1**   For assessing salient information, we adopted token-embedding-based metric BERTSCORE-F1. We used the pre-trained 'microsoft/deberta-xlarge-mnli' version provided

by (Zhang* et al., 2020) as the state-of-the-art checkpoint.

## A.1 Human Evaluation

We conduct A/B testing to evaluate if our proposed method could actually aid in reducing framing bias. We compare generations from the model with our proposed polarity minimization loss $L_{polar}$ – LR-INFO, against the generations from the state-of-the-art model, a BART-large model fine-tuned on ALL-SIDES dataset **without** the loss (BARTNEUSFT-T). We got the generation by running the publicly available checkpoint from BARTNEUSFT-T (Lee et al., 2022).

We conducted the evaluation with 30 randomly selected samples. We provide two articles from the two models (in random order) along with the issue sentence that describes what the articles are about. Then, the annotator is asked to answer the question "Which article is more biased?", following Spinde et al. (2021); Lee et al. (2022). We get three annotations for each sample and select the majority voting. Since many of the test samples are closely related to U.S. politics, we recruited three **non-**U.S. citizens/nationals/residents to minimize any political bias or personal preference involved in the evaluation. All three annotators claimed themselves as moderate in political leaning and they are qualified to conduct the evaluation in English (they all have received their tertiary education in English).

To verify that the selection of which one is biased in the pairs is not random, a binomial test is conducted after obtaining the evaluation results. The null hypothesis was "The selection of articles generated from LR-INFO (our proposed method) to be less biased is random". Then, we obtained a p-value of 0.019, which rejected the null hypothesis ($p < 0.05$). Therefore, the selection of articles generated from LR-INFO to be less biased is not random.

When the model is trained with polarity minimization loss, it can learn to remove bias-inducing information while BARTNEUSFT-T suffers. As illustrated in Table 4, our model LR-INFO could remove bias-inducing information "Trump is expected to attack President Joe Biden's immigration policies" from the summary about the issue of "Trump to speak at CPAC" while BARTNEUSFT-T failed to remove it.

| |
|---|
| **Issue**: Trump to Speak at CPAC |
| [LR-INFO] Former President Donald Trump will address the Conservative Political Action Conference (CPAC) on Sunday, his first public speaking engagement since leaving office. |
| [BARTNEUSFT-T ] Former President Donald Trump will give a keynote address at the Conservative Political Action Conference (CPAC) on Sunday, his first speaking engagement since leaving office. **Trump is expected to attack President Joe Biden's immigration policies.** |

Table 4: Human Evaluation Example.

## A.2 Full Experimental Result over varying weights of $L_{polar}$

We investigated the effect of the weights of the polarity minimization loss $L_{polar}$ with respect to $L_{MDS}$ in the range of $\lambda \in \{0.3, 0.7, 1, 1.5\}$. Table 5 shows the full result for pour proposed models with polarity minimization losses with varying weights $\lambda$.

| | Weight ($\lambda$) | Avg. Framing Bias Metric $\downarrow$ | | | Salient Info $\uparrow$ | |
|---|---|---|---|---|---|---|
| | | $Arousal_+$ | $Arousal_-$ | $Arousal_{sum}$ | BLEU | BERTSCORE-F1 |
| LR-VALENCE | 0.3 | 1.583 | 0.91 | 2.493 | 10.9 | 69.96% |
| | 0.7 | 1.688 | 0.919 | 2.607 | 10.86 | 69.10% |
| | 1 | 1.572 | 0.913 | 2.486 | 10.62 | 69.67% |
| | 1.5 | 1.48 | 0.856 | 2.336 | 10.9 | 69.83% |
| LR-AROUSAL | 0.3 | 1.342 | 0.788 | 2.13 | 10.23 | 70.00% |
| | 0.7 | 1.468 | 0.876 | 2.345 | 11.13 | 70.19% |
| | 1 | 1.183 | 0.615 | 1.798 | 8.84 | 69.49% |
| | 1.5 | 1.341 | 0.831 | 2.171 | 10.92 | 70.56% |
| LR-INFO | 0.3 | 1.347 | 0.791 | 2.138 | 10.24 | 69.81% |
| | 0.7 | 1.081 | 0.698 | 1.778 | 9.31 | 70.20% |
| | 1 | 1.436 | 0.775 | 2.212 | 10.44 | 70.13% |
| | 1.5 | 1.529 | 0.86 | 2.388 | 10.6 | 69.60% |
| LRC-AROUSAL | 0.3 | 1.664 | 0.962 | 2.626 | 11.4 | 69.85% |
| | 0.7 | 1.412 | 0.834 | 2.246 | 10.65 | 69.93% |
| | 1 | 1.218 | 0.747 | 1.966 | 9.66 | 69.94% |
| | 1.5 | 1.536 | 0.87 | 2.406 | 10.88 | 70.10% |
| LRC-INFO | 0.3 | 1.413 | 0.844 | 2.257 | 11.23 | 70.90% |
| | 0.7 | 1.248 | 0.722 | 1.97 | 10.18 | 70.24% |
| | 1 | 1.369 | 0.857 | 2.226 | 10.73 | 70.48% |
| | 1.5 | 1.352 | 0.789 | 2.141 | 11.78 | 70.74% |

Table 5: Experimental results for our models with proposed polarity minimization loss, LR-VALENCE, LR-AROUSAL, LR-INFO, LRC-AROUSAL, LRC-INFO, with varying weights ($\lambda$). For framing bias metric, the *lower* number is the better ($\downarrow$). For other scores, the *higher* number is the better ($\uparrow$). The results of our models with polarity minimization loss (those denote with +) are reported with best $\lambda$. Full exploration of $\lambda$ is available in Appendix and Fig. 2

# B Generation Results

In Table 7, 6, We provide examples of generations from our model LR-INFO and LRC-AROUSAL for better understanding of the effectiveness of our proposed polarity minimization loss $L_{polar}$. Additionally, we also provide the corresponding example generations from ablation study described in Subsection 4.3.

| Issue | *Trump Returns to Campaigning; Doctor Says He is 'No Longer Contagious'* |
|---|---|
| Inputs | $\mathbf{X^L}$ : Trump returns to public campaigning, falsely claiming that the virus that infected him is 'disappearing'
$\mathbf{X^R}$ : Trump no longer at risk of spreading COVID-19, doctor says
$\mathbf{X^C}$ : Doctor Says Trump Isn't Transmission Risk After President Holds Public Event |
| Generations | **[R->L]** Trump Holds First Public Event Since Covid-19.
**[L->R]** Trump Holds First Public Event Since Covid-19.
**[LRC-AROUSAL]** Trump Returns to Public Campaigning.
**[LR-INFO (best)]** Trump Holds First Public Event Since Contracting COVID-19. |
| **Issue** | *Mueller Issues Indictments Against Russians* |
| Inputs | $\mathbf{X^L}$ : What Mueller's new Russia indictments mean — and don't mean
$\mathbf{X^R}$ : Latest Mueller indictment complicates Russian collusion narrative
$\mathbf{X^C}$ : Five key takeaways from the Russian indictments |
| Generations | **[R->L]** Mueller Indicts 13 Russians.
**[L->R]** New Russia Indictments.
**[LRC-AROUSAL]** Mueller Indicts 13 Russians.
**[LR-INFO (best)]** Mueller Indicts 13 Russians. |
| **Issue** | *What to Watch for: 2020 Gubernatorial Races* |
| Inputs | $\mathbf{X^L}$ : The 3 biggest governor races to watch in 2020
$\mathbf{X^R}$ : Meet the Highest-Polling Libertarian Gubernatorial Candidate in the Country
$\mathbf{X^C}$ : 11 States Are Choosing Their Governor On Tuesday. Here Are Races To Watch |
| Generations | **[R->L]** 11 States Are Choosing Their Next Governor.
**[L->R]** Governor Races to Watch in 2020.
**[LRC-AROUSAL]** Governor Races to Watch in 2020.
**[LR-INFO (best)]** 11 States Choose Their Next Governor. |
| **Issue** | *What to Watch for: 2020 Gubernatorial Races* |
| Inputs | $\mathbf{X^L}$ : The 3 biggest governor races to watch in 2020
$\mathbf{X^R}$ : Meet the Highest-Polling Libertarian Gubernatorial Candidate in the Country
$\mathbf{X^C}$ : 11 States Are Choosing Their Governor On Tuesday. Here Are Races To Watch |
| Generations | **[R->L]** 11 States Are Choosing Their Next Governor.
**[L->R]** Governor Races to Watch in 2020.
**[LRC-AROUSAL]** Governor Races to Watch in 2020.
**[LR-INFO (best)]** 11 States Choose Their Next Governor. |

Table 6: Illustration of example generations. $X^L, X^R, X^C$ denote polarized input articles from left-, right-, center-leaning media outlets. LR-INFO is generation from the best model with our proposed polarity minimization loss $L_{\text{polar}}$. A->B denotes when the model learns one direction of polarization from A to B, which is equivalent to $-$"$BtoA$" in our ablation study.

| Issue | **2018 Midterms: Will There Be a Blue Wave?** |
|-------|-----------------------------------------------|
| Inputs | $X^L$ : Get Over Your Election-Needle P.T.S.D.: The Blue Wave Is Real, and It's a Monster
$X^R$ : Don't Get Too Excited about Election Day Yet, Democrats
$X^C$ : Midwest Abandons Trump, Fueling Democratic Advantage For Control Of Congress |
| Generations | **[R->L]** The Blue Wave Is Real, and It's a Monster.
**[L->R]** Midwesterners Are More Likely to Vote for Democrats.
**[LRC-AROUSAL]** Polls Suggests Democrats Will Win Control of Congress.
**[LR-INFO (best)]** Midterm elections in the U.S. Election. |

| Issue | **Off-Year State Elections Prompt 2020 Speculation** |
|-------|------------------------------------------------------|
| Inputs | $X^L$ : Polls close in Kentucky, Mississippi to follow, in elections testing Trump's political power
$X^R$ : GOP's Bevin trailing in Kentucky gubernatorial race, as Trump calls for 'angry majority' to rise
$X^C$ : Polls close as off-year election results offer clues to 2020 |
| Generations | **[R->L]** Polls Close in Kentucky, Mississippi, New Jersey and Virginia.
**[L->R]** Polls Close in Kentucky, Mississippi, New Jersey and Virginia.
**[LRC-AROUSAL]** Polls Close in Kentucky, Mississippi, New Jersey and Virginia Governor's Races.
**[LR-INFO (best)]** Off-Year Midterm Elections in Kentucky, Mississippi, New Jersey and Virginia Offer Clues to 2020. |

| Issue | **GOP Lawmakers Consider Gun Control** |
|-------|----------------------------------------|
| Inputs | $X^L$ : Some in GOP open to discussing Democrats' proposal to ban device used in Las Vegas attack
$X^R$ : Republicans Get Behind Gun Control in Wake of Las Vegas Shooting
$X^C$ : GOP Lawmakers Consider Gun-Control Measure |
| Generations | **[R->L]** Republicans Are Open to Banning Bump Stocks.
**[L->R]** GOP Lawmakers Consider Gun-Control Measure.
**[LRC-AROUSAL]** Republicans Consider Gun Control in Wake of Las Vegas Shooting.
**[LR-INFO (best)]** GOP Lawmakers Consider Gun Control Measures in Wake of Las Vegas Shooting. |

Table 7: Illustration of example generations 2. $X^L, X^R, X^C$ denote polarized input articles from left-, right-, center-leaning media outlets. LR-INFO is generation from the best model with our proposed polarity minimization loss $L_{\text{polar}}$. A->B denotes when the model learns one direction of polarization from A to B, which is equivalent to $-$"$BtoA$" in our ablation study.