# OpenReview forum: "Mitigating Framing Bias with Polarity Minimization Loss"
_EMNLP/2023/Conference — EMNLP 2023 Findings_

### Official Review · Reviewer_rWMe · 2023-07-21

**Soundness:** 3

**Excitement:**

3: Ambivalent: It has merits (e.g., it reports state-of-the-art results, the idea is nice), but there are key weaknesses (e.g., it describes incremental work), and it can significantly benefit from another round of revision. However, I won't object to accepting it if my co-reviewers champion it.

**Missing References:**

None that I know of, besides my point above about choice of baselines.

**Paper Topic And Main Contributions:**

The authors present the case for the need for depolarising news summarisation to improve access to neutral news information. They propose a loss function with two components; the first aims to generate a summary article conditioned on polarised source articles; the second aims to optimise the (bi-directional) mapping between opposed articles (in order to identify the “common ground”). The authors show variations of their methodology outperform general-purpose summarisation models with regards to neutralising valence and retaining salient information.  They also demonstrate that their methodology can outperform a summariser finetuned on sets of polarised articles and neutral summaries, with regards to subjective polarity, suggesting their technique is effective.


**Questions For The Authors:**

(A) Why did you compare the model only to general-purpose summarisation models rather than to the other neutralisation model for the benchmarks, given yours is not the first? Why did you conduct human evaluation compared to a model you do not provide benchmark results for?

**Reasons To Accept:**

The paper is well motivated with a very clear introduction, setting out the problem and introducing key concepts to readers unfamiliar with the topic.

The authors use existing benchmarks to measure improved neutrality and key information retention, and supplement this with a human evaluation that demonstrates that perceived polarity has been reduced. Showing improvements across a range of metrics supports their claims of the effectiveness of their approach. The ablation experiments provide further evidence for their claim that learning bidirectional polarity shifts are crucial for achieving neutrality.



**Reasons To Reject:**

A concern I have is the selective comparison to general-purpose summarisation models. In Related Works in Appendix A, the authors mention a model produced by Lee et al. (2022) -  but give no reason why this is not considered as a baseline for comparison. Another model from Lee at al. (2022) (BARTfinetuned) is used for human evaluation, but not for the benchmark comparison. A glance at this paper suggests comparable results, if not improved, so the reason for omissions is unclear.


**Reproducibility:**

5: Could easily reproduce the results.

**Reviewer Confidence:**

2: Willing to defend my evaluation, but it is fairly likely that I missed some details, didn't understand some central points, or can't be sure about the novelty of the work.

**Typos Grammar Style And Presentation Improvements:**

Section 2 - I think this section would benefit from some accompanying “plain English” to make the purpose of each element of the loss function clearer.

Line 095 - erroneous capitalisation

Line 203 - quite much → relatively well

Line 211 - I found the paragraph “Effective learning with extreme polarities” hard to follow. The first sentence - “we could investigate” - could imply that you did not then investigate this topic, which disrupts the flow a little. The main argument seems to be that learning is effective from extreme polarities, but one of the examples you give is the LRC-arousal model which has a three-way comparison.

Line 515 - citations given with no date.

Figure 2 - as this is not referenced in the discussion, I would suggest moving this to the Appendix to give yourself more room in the main body of the paper if needed.

---

> ### Author Rebuttal · Authors · 2023-08-29
>
> We appreciate your constructive feedback and will improve the writing based on your feedback.
>
> To answer your question, we compared with general-purpose summarization models to show the importance of polarization minimization loss in removing framing bias. Following your suggestion, we have added the comparison to the other neutralization model as follows:
>
> |  Model  |  Arousal+  | Arousal- | Arousal_sum | BLEU | Bertscore |
> |--|:--:|:--:|:--:|:--:|:--:|
> |  BART-finetuned    | 1.86   | 1.00   |  2.85  |  11.57  |  70.10%  |
>
> As advised, we will add the results from the benchmark for the complete comparison in the final manuscript with an additional page.

---

### Official Review · Reviewer_xt1k · 2023-07-23

**Soundness:** 3

**Excitement:**

3: Ambivalent: It has merits (e.g., it reports state-of-the-art results, the idea is nice), but there are key weaknesses (e.g., it describes incremental work), and it can significantly benefit from another round of revision. However, I won't object to accepting it if my co-reviewers champion it.

**Missing References:**

[1] In Plain Sight: Media Bias Through the Lens of Factual Reporting

[2] Context in Informational Bias Detection

[3] We Can Detect Your Bias: Predicting the Political Ideology of News Articles

[4] POLITICS: Pretraining with Same-story Article Comparison for Ideology Prediction and Stance Detection

[5] PRIMERA: Pyramid-based Masked Sentence Pre-training for Multi-document Summarization

**Paper Topic And Main Contributions:**

This paper identify the framing bias issue in media outlets such that different media outlets have a skewed selection of language and information when reporting on the same news event. Inspired by lexical and informational bias in Framing Bias, the author proposes Polarity Regularized Model by designing Polarity minimization Loss which is to train a summarization model to generate articles from one end of the political spectrum to another end (i.e., generating an article of opposite ideology). The author conducts experiments to show the effectiveness of their model by evaluating on two aspects: bias mitigation and information preservation.

**Questions For The Authors:**

Following weakness 1 about the intuition, could you elaborate more on this point? As pointed out in line 039-045, you seem to prime the model to focus on the common ground by forcing the model to generate both same-side and opposite opinions. In this case, would it make more sense to compute losses only on the "common phrases/words/sequences" instead of the full sequence of opposite opinions?

I don't quite follow how you construct {X_S, X_T} (line 177-192). For example, 1) for the LRC-arousal approach, what do you mean by "compose the set with tuples of the highest arousal score and the lowest arousal score"? My imagination is that, similar to LR-arousal, you would first pick a pair of L and R articles that have a high difference in valence score, but then how do you include the C article? Also, why do you want to include tuples that have the lowest arousal score? 2) for the informational approach, what do you mean by "high differences between the numbers of unique tokens"? For example, which pair would you choose: pair 1: L and R each has 100 unique tokens, and pair 2: L and R each has 10 unique tokens? In both cases, the difference is 0 but I guess you would choose pair 1 since more "information bias" is observed. Similarly, how do you include C article.

From my reading of your manuscript, it's unclear to me whether PEGASUSMULTI and BARTMULTI are fine-tuned on AllSides since you said "compare with off-the-shelf multi-document summarization models" (line 155-156). Please correct me if I am wrong. However, if not fine-tuned, it's advised to include results for fine-tuned BART and PEGASUS. Meanwhile, which underlying model are you using when you report "+LR" results in the bottom panel in Table 1.

**Reasons To Accept:**

The motivation behind "polarity minimization loss" is interesting which can boil down to the two aspects of framing bias: lexical and information bias (though it would be good to cite some recent NLP working along this line, e.g., "In Plain Sight: Media Bias Through the Lens of Factual Reporting", "Context in Informational Bias Detection").

The proposed method outperforms all baselines in terms of both Framing Bias metric (assessing how much bias is reduced) and Salient Info metric (assessing how much info is preserved in the generated summary, e.g., BLEU).

Interesting ablation study: Polarity shift in uni-direction. This ablation study shows that it's important to consider both directions in framing bias mitigation, and the sample generation in Table 2 well echoes this point. It would be nice to see more examples of such kind.

**Reasons To Reject:**

The intuition behind the proposed "polarity minimization loss" is not well articulated. Similar "polarity minimization" idea is seen in [3][4], though they are not working on summarization tasks, where they train the model to minimize the representation difference between two articles of similar ideology but maximize between articles of opposite ideologies. Their approaches seem to be more **explicit** by forcing the model to directly learn the common ground; whereas this paper seems to **indirectly** learn the common ground by forcing the model to generate R article using L article, which doesn't seem to be sound. Also, see "Questions For The Authors" where I suggest another more explicit way to learn the common ground, which should have been included as a baseline.

Experimental design can be improved: 1) include more recent multi-doc summarization technique (e.g., PRIMERA), 2) include fine-tuned BART and PEGASUS (see details in Questions For The Authors)

Some texts are hard to follow (e.g., line 177-192), and see details in Questions For The Authors. The way the author constructs the pairs/tuples for polarity minimization loss is unclear, which makes it hard to judge the soundness of the model design.

Writing can be improved since there are some minor typos (see Typos Grammar Style And Presentation Improvements).

**Reproducibility:**

4: Could mostly reproduce the results, but there may be some variation because of sample variance or minor variations in their interpretation of the protocol or method.

**Reviewer Confidence:**

4: Quite sure. I tried to check the important points carefully. It's unlikely, though conceivable, that I missed something that should affect my ratings.

**Typos Grammar Style And Presentation Improvements:**

Line 043: "in Fig 1" not "Fig 2"

Line 77: with polarized articles (no "a")

Line 83: "defined in eq. 1"

Line 116: I suppose you mean "bound to"

Line 124: remove the comma at the beginning.

If accepted, please move Appendix A (related work) to the main content, and include missing references mentioned above as well as discuss the similarities/differences between your work and existing work.

---

> ### Author Rebuttal · Authors · 2023-08-29
>
> We appreciate your constructive feedback and your effort. We would like to answer your questions and clarify some misunderstandings as below.
>
> **1. About "polarity minimization loss"**
>
> Polarity minimization loss is for explicitly teaching how politically polarized articles are mapped to each other in both directions (e.g., both L→ R, and R→ L simultaneously), thus the model can learn how an issue is represented in opposite polarities. Since our  L_polar allows mapping from each other, it will ultimately force the model to learn to generate the common ground (“common phrases/words/sequences”) between two extreme political views (i.e., polarized ends).
>
> To respectfully explain the distinction of our work, the previous works [3,4] focus on ideology prediction (if they are left-, right-, or center-leaning) or stance prediction -- which is polarity detection. On the other hand, our work focuses on **generating** a neural article from polarized articles. Although it is an important task to detect political ideology (classification) [3,4], it is also important to generate a neutral summary (generation) from political ideologies to mitigate polarization. Thank you for introducing [3,4] works. As advised, we will move the Related Work section to the main body with an additional page and similarities/differences between our work and existing work, including [3,4] (political ideology/stance detection works).
>
> [3] We Can Detect Your Bias: Predicting the Political Ideology of News Articles (Baly et al., EMNLP 2020)
>
> [4] POLITICS: Pretraining with Same-story Article Comparison for Ideology Prediction and Stance Detection(Liu et al., Findings 2022)
>
> **2. Construction of {X_S, X_T} (line 177-192)**
>
> We would like to clarify the misunderstanding caused by our writing as follows with some examples:
>
> - **2-1) LRC-Arousal:** Your understanding about LR-arousal is correct. Similarly, the same logic applies to LRC-Arousal. Yet, unlike LR-arousal, we utilize all three of X^L, X^R, X^C source articles (Here, X^C is also a biased article as explained in lines 135-137). In other words, to pick pairs of articles that have high differences in arousal score, we calculate differences of all combinations, which are {L, R}, {L, C}, {R, C} whereas LR-only calculator {L, R}.
>
> - **2-2) Informational approach:** We counted the number of unique tokens between two pairs and selected those that have the highest count combined. We should modify our writing to “highest difference in information based on the sum of unique tokens from each article from the pair.”
>
> **3. Data for Baselines & Additional Baselines**
>
> - **3-1) PEGASUSMULTI, BARTMULTI baselines:** are finetuned on the multi-news dataset (Fabbri et al., 2019) as stated in line 159, thus, we called it off-the-shelf models.
>
> -  **3-2) Results of fine-tuned BART & PEGASUS:** As advised, we would like to share the results of the fine-tuned BART, and fine-tuned PEGASUS as follows:
> |  Model  |  Arousal+  | Arousal- | Arousal_sum | BLEU | Bertscore |
> |--|:--:|:--:|:--:|:--:|:--:|
> |  BART-finetuned    | 1.86   | 1.00   |  2.85  |  11.57  |  70.10%  |
> |  PEGASUS-finetuned  |  2.18   |  1.12  |  3.3   | 11.26   |  68.41%  |
> | + LR-VALENCE (Ours)  | 1.57 |  0.91 |  2.48 |  10.62 |  69.67% |
> | + LR-AROUSAL (Ours)  |  1.18 |  0.62 | 1.80 |  8.84 |  69.49% |
> | + LR-INFO (Ours)  |  1.08 |  0.70 |  1.78 |  9.31 |  70.19% |
> We will update our manuscript with the results.
>
> - **3-3) the base model for Table 1:** We utilize the BART-large model for Table 1. We will remove the “+” sign to remove the confusion, instead, we will add BART_{LR-XYZ} to make it clear about the base model.
>
> - **3-4) PRIMERA result:**
> Additionally, as suggested by the reviewer, we also share the suggested the result on PRIMERA model:
> |  Model  |  Arousal+  | Arousal- | Arousal_sum | BLEU | Bertscore |
> |--|:--:|:--:|:--:|:--:|:--:|
> |  PRIMERA    | 3.99   | 1.95   |  5.94  |  8.31  |  65.32%  |
>
> [Reference]
> Multi-News: A Large-Scale Multi-Document Summarization Dataset and Abstractive Hierarchical Model (Fabbri et al., ACL 2019)

---

### Official Review · Reviewer_X96S · 2023-08-05

**Typos Grammar Style And Presentation Improvements:** 1. In footnote 1, the index of the Ap…
**Soundness:** 3

**Excitement:**

3: Ambivalent: It has merits (e.g., it reports state-of-the-art results, the idea is nice), but there are key weaknesses (e.g., it describes incremental work), and it can significantly benefit from another round of revision. However, I won't object to accepting it if my co-reviewers champion it.

**Paper Topic And Main Contributions:**

This paper proposes a polarity minimization loss to reduce the framing bias in the generation. Their experiment results show the effectiveness of the addition of the proposed loss.

**Questions For The Authors:**

See above in Reasons To Reject

**Reasons To Accept:**

1. The authors propose a polarity minimization loss incorporating into the causalLM loss and show the benefits of addressing framing bias.
2. They provide a complete experiment, including the comparison of the baseline methods, ablation study, and generation analysis.
3. The paper structure is clear and it is easy to follow.


**Reasons To Reject:**

1. More explanation in the experiment part is needed.
1.1. What is the "All Source" model in Table 1? Is the baseline model trained on the ALLSIDE dataset? And what is the model used here?
1.2. Similarly, In Table 1, what is the model used in the experiments of "+LR..."? Is it adopted from "All Source"  or "BARTMULTI"?
1.3. What does the "Arousal+/-" mean? Assuming it is corresponding to the party left and right. But which one corresponds to which?
2. The discussion of Figure 2 is confusing to me.
2.1. The only part that mentions Fig 2 is in line 043 but that would be Figure 1 instead?
2.2. And resultly, what is the selection of \lambda for the experiment in Table 1 "+LR..."? Would that be the best for every setting or the same setting(\lambda) for every experiment?
2.3. Also, What is causing the variant in the selection of \lambda? Could you please provide some analysis on that?

3. I'm slightly skeptical about the IAA in line 235 for human evaluation. Could you please also provide the result of the binomial test showing that the selection of which one is biased in the pairs is not random?

**Reproducibility:**

3: Could reproduce the results with some difficulty. The settings of parameters are underspecified or subjectively determined; the training/evaluation data are not widely available.

**Reviewer Confidence:**

4: Quite sure. I tried to check the important points carefully. It's unlikely, though conceivable, that I missed something that should affect my ratings.

---

> ### Author Rebuttal · Authors · 2023-08-29
>
> We appreciate the effort you have taken to provide feedback. Please find our answers and clarification to your questions below:
>
> **1. Clarification on names of baseline proposed models and metrics:**
>
> - **1.1 All Source:** All Source is **not** a result of generation from a model, but evaluation on the concatenation of all input sources (X^L, X^R, X^C). We report it to show a lower bound for framing bias metric as we assumed that source input contains bias. Also, it can indicate an idea of the upper bound for salient information. We will update it as “All source input” to remove confusion caused by its name.
>
> - **1.2 What is +LR-XYZ?** Those are **not** based on either of the All Source or BARTMULTI. All experiments with +LR(XYZ) are done with the BART-base model using the Allsides dataset. We named each model “+” to show how L_polar (polarity minimization term) is calculated on top of the multi-document summarization loss L_MDS.  We acknowledge that it causes misunderstanding, thus we will remove the + sign and also make it clear experimental results are based on the BART-base model with the Allside dataset.
>
> - **1.3 What does "Arousal+/-" mean?** They are not related to political leanings but they are based on valence (positive or negative) based on VAD lexicons (Mohammad, 2018). As defined by Lee et al., (2022), Arousal + / - stands for positive-valence and negative-valence arousal scores respectively. For instance, a word “waste” has a high arousal score with positive valence.
>
> **2. Regarding Figure 2**
>
> Thank you for pointing out. We will add the following clarification about Figure 2 (regarding $\lambda$) in the final manuscript.
>
> - **2.1. Line 43:** Yes, line 043 should be referring to Figure 1. Figure 2. Illustrates the effect of $\lambda$ (weight of polarity minimization loss L_polar) on framing bias scores. We will clarify them in the paper.
>
> - **2.2. Regarding $\lambda$:** In Table 1, we reported the best setting for each model. For completeness, we shared all results for all lambda variants we tried in Appendix and Figure 2. In the final version, we will explicitly state “The results are reported with best $\lambda$.” in the caption of Table 1.
>
> - **2.3. Analysis on a variant of $\lambda$:** $\lambda$ is a hyperparameter that assigns weight to the polarity minimization loss (L_polar) with respect to the multi-document summarization loss (L_MDS). $\lambda$ also indicates the intensity of the signal for learning opposite polarity. The framing bias scores increase when polarity minimization loss overtakes MDS loss (i.e.,  $\lambda$ == 1.5) for most of the models, except for +LR-Valence model. In general, models remove framing bias most effectively when L_polar weighs the same ($\lambda$=1) or slightly less ($\lambda$=0.7) in respect to L_MDS. The variant indicates there exist optimal weights for each model instead of a universal optimal weight.
>
> **3. Binomial test on human annotation:**
>
> We conducted a binomial test with a null hypothesis “The selection of articles generated from LR-INFO (proposed method) to be less biased is random”. Then, we obtained a p-value of 0.019, which rejected the null hypothesis (p < 0.05). Therefore, the selection of articles generated from LR-INFO to be less biased is not random.

---

### Meta-Review · Area_Chair_Ua4j · 2023-09-15

**Recommendation:** 3

**Metareview:**

This paper presents a method for obtaining neutral articles from a multi-document summarization system by introducing a polarity minimization loss into the overall loss.

Like at least one of the reviewers, I found the intuition for the specific loss function neither well-described nor well-motivated, nor did I find the authors' rebuttal helpful in further elucidating this part.  This is the crux of the work, so this was rather important.

Nevertheless, this paper still presents a worthwhile problem with a novel loss function and good—if sometimes hard-to-follow—analysis.

---

### Decision · Program_Chairs · 2023-10-07

**Decision:**

Accept-Findings

**Comment:**

This paper presents a method for obtaining neutral articles from a multi-document summarization system by introducing a polarity minimization loss into the overall loss.

Like at least one of the reviewers, I found the intuition for the specific loss function neither well-described nor well-motivated, nor did I find the authors' rebuttal helpful in further elucidating this part.  This is the crux of the work, so this was rather important.

Nevertheless, this paper still presents a worthwhile problem with a novel loss function and good—if sometimes hard-to-follow—analysis.